# Frontline Healthcare Professionals’ Views Regarding the Impact of COVID-19 on Ethical Decision-Making: A Multicentre Mixed-Methods Study from Estonia

**DOI:** 10.3390/healthcare10040711

**Published:** 2022-04-12

**Authors:** Kadri Simm, Jay Zameska, Kadi Lubi

**Affiliations:** 1Department of Philosophy, Institute of Philosophy and Semiotics, University of Tartu, 50090 Tartu, Estonia; 2Tallinn Healthcare College, Tallinn University of Technology, 19086 Tallinn, Estonia; jay.allen.zameska@ut.ee (J.Z.); kadi.lubi@taltech.ee (K.L.)

**Keywords:** COVID-19, ethical decision-making, triage, frontline healthcare workers, mental health, Estonia, pandemic

## Abstract

Background: The objective of the study was to investigate frontline healthcare professionals’ experiences and attitudes in relation to the COVID-19 pandemic’s ethical and psychosocial aspects in Estonia. There were two research foci: first, ethical decision-making related to treating patients in the context of potential medical resource scarcity, and second, other psychosocial factors for healthcare professionals pertaining to coping, role conflicts, and the availability of institutional support. Methods: An online survey was conducted in the fall of 2020 amongst the frontline healthcare professionals working in the three most impacted hospitals; respondents were also drawn from two ambulance services. The focus of the survey was on the first wave of COVID-19 (spring 2020). A total of 215 respondents completed the quantitative survey and qualitative data were gathered from open comments. Results: Over half of the surveyed healthcare professionals in Estonia expressed confidence in their roles during the pandemic. More than half cited the complex ethical aspects related to their decisions as their main source of doubt and uncertainty. In response to this uncertainty, Estonian healthcare professionals drew on their previous training and experience, the policies and guidelines of their institution, and support from their colleagues, to aid their decision-making during the pandemic. Conclusions: Although frontline healthcare professionals faced difficult decisions during the first wave of the pandemic, overall, most agreed that experiencing the pandemic reconfirmed that their work mattered greatly.

## 1. Introduction

The COVID-19 pandemic, especially in its first wave in the early spring of 2020, brought along many new challenges for frontline healthcare professionals (HCP). Various aspects of patient management changed—there were modifications to admission strategies and triage principles, and to rules regarding the transfer of patients between units and keeping a physical distance from the patient. These uncertain times have had an effect on HCP, and numerous ethical and psychosocial dilemmas have already been discussed in the literature; this includes both theoretical accounts regarding the normative grounds for resource allocation [1,2,3,4], as well as empirical overviews that have mapped the attitudes and experiences of HCP during the pandemic, especially in relation to their mental health [5,6,7,8,9]. Previous research has shown that although the first wave (the focus of the present article) was mild in the Baltics, witnessing the dramatic scenes in China, Italy, and Spain accelerated the establishment of pandemic policies at a national level [10]. Thus, while a health care crisis did not actualize here during the first wave, anticipation and “bracing oneself” for the arrival of the pandemic resulted in rapid modifications to patient management systems and changes to the mental preparedness of the frontline health care workforce that we aimed to investigate.

Standardized questionnaires have been widely used during the COVID-19 crisis to measure changes in mental health amongst populations of HCP, with the results indicating that direct contact with COVID-19 patients and working at high-risk hospitals are the main factors in raising the risks of mental stress (burnout, depression, anxiety, quality of sleep, etc.) [11]. The objectives of our study, which were exploratory and scoping in nature, rather than measuring, were to broaden the investigation of the experiences and attitudes of frontline HCP in relation to the COVID-19 pandemic’s ethical and psychosocial aspects in Estonia. There were two research areas: first, ethical decision-making related to treating patients in the context of potential medical resource scarcity (confidence and sources of support), and second, other psychosocial factors for HCP pertaining to coping, role conflicts (professional ethics vs. duties to their families), and the availability of institutional support.

## 2. Methods

### 2.1. Study Site

The study site was Estonia (population 1.3 M) and the focus was the first pandemic wave during the spring of 2020. In hindsight, the 1st wave turned out to be relatively mild: infection rates peaked at 57 persons per 100,000 inhabitants in April 2020, compared with 1553 persons per 100,000 at the peak of the 2nd wave in mid-March 2021, and 1817 at the peak of the 3rd wave in November 2021 [12]. Yet, the news from battered Italian and Spanish hospitals very much set the mood and fed the anxieties of the medical community, as well as wider society, during the first wave of spring 2020.

During the first wave, the spread of coronavirus in Estonia was most extensive on the island of Saaremaa. In mainland Estonia, the COVID-19 patients needing hospitalization were mostly admitted to the country’s two largest hospitals. Our study sample thus included the following three hospitals (out of a total of 19 in Estonia): Kuressaare hospital on Saaremaa Island, North-Estonia Regional Hospital in the capital Tallinn, and Tartu University Hospital in the second largest city, Tartu, located in southern Estonia. In addition, our sample included two ambulance services (out of a total of ten) in the two largest cities in Estonia (Tallinn and Tartu).

### 2.2. Data Collection and Participants

A purposeful sampling method was used: we sent a recruitment and information email to the heads of the hospital administration and emergency services, who then forwarded the survey link to their employees who fit the recruitment profile. For the purposes of the study, front-line work was defined as an employment situation where one has a high risk of coming into contact (or where one actually comes into contact) with COVID-19-infected persons. The target group for the study sample had the following inclusion criteria: working in an ICU, ER, or ambulance department of the selected institutions at the time of the first wave of the pandemic. A total of 215 respondents completed the survey during the data collection period from 16 October to 30 November 2020 (November 2020 coincided with the slow beginning of the second wave in Estonia). No particular restrictions were in place during data collection, and it was carried out electronically.

Data were collected through a quantitative online questionnaire (survey platform survey.ut.ee of the University of Tartu was used). The questionnaire contained 13 closed-ended questions, and a section for background variables such as age, specialty, duration of working experience, and working unit. Qualitative insights were gained through free text comment boxes (N = 208) that were open for 8 questions (these comment boxes offered the opportunity to comment on, or explain, one’s choice of “other”). The qualitative elements of the survey were analyzed with the aim of illustrating, and elaborating on, the most significant findings of the quantitative section. Quantitative data were analyzed using MS Excel descriptive statistics and Bluesky Statistics version 7, with frequencies given in percentages, and *p* ≤ 0.05 was considered statistically significant.

## 3. Results

Participant age groups and specialties are described in Table 1.

Our results showed that 35% of respondents worked in ICU and ER settings, 60.5% in ambulance settings, and numerous respondents worked in multiple settings. Responses to the two linked questions about (self-)confidence are shown in Table 2 (Q1 was “how (self-)confident were you in your clinical role during the pandemic”, and Q2 related to (self-)confidence in cases where access to medical resources such as ventilators would need to be limited). Notably, EMTs (emergency medical technicians) offered the most varied assessments of their confidence, with a statistically significant relationship between role and confidence (*p* = 0.0367). For the second, more critical, question, the confidence levels of all frontline HCP predictably decreased. During the first wave of the pandemic in Estonia, the need for such difficult allocation decisions never materialized, although the harrowing reports from China and Northern Italy at the time likely affected people. We found that 86% of respondents fully or mostly agreed that it was their professional training that allowed them to proceed with medically and ethically correct decisions, while 10% did not feel that way (fully or mostly disagreed).

We were interested in finding out what HCP themselves saw as meaningful and useful actions that would help to ensure that they would be better prepared in the future. Table 3 outlines the most significant sources of support for medical decision-making during the pandemic.

In the open commentary section, several respondents highlighted opportunities offered by telemedicine and consultation, as well as online research articles and results, that offered significant support for medical decision-making. In addition, we also asked the participants to identify sources of general support during the crisis (up to three choices). Not surprisingly, family and friends played an important role (86%), as did close colleagues (84%), and the institution itself (64%). Responses were more mixed in terms of other factors, for example 38% sometimes experienced the supportive role of government and politicians, but 27% of respondents disagreed with this; 32% sometimes felt the support of society more widely, yet 24% did not. The main reasons for doubts and uncertainty, amongst those who felt them, were the complex ethical aspects related to their decisions (63%), the ambiguities and opacity of the decision-making process (39%), lack of experience (38%), and lack of knowledge (28%) (multiple replies were allowed).

Table 4 describes frontline HCP views on what criteria, in the context of resource scarcity, should form the basis for treatment decisions.The first two criteria—expected outcome of treatment (43.1%) and the health status of the patient (32.8%)—are often (although not always) linked in medical context, and were judged to be more significant than the other criteria. The specificity of COVID-19 made the patient’s age the third most relevant criteria, as the elderly are the highest-risk group and also the biggest users of medical resources. The rest of the listed criteria (privileging of medical workers, patient’s disability, and treatment on a first-come-first-served basis) were rarely selected, thus their importance to front line medical workers was low. The results showed that 83% of respondents held the view that decisions to limit treatment should be made by a team including the responsible treating physician.

We also asked whether, and how, pandemic triage differed from ordinary triage. Responses were almost evenly split amongst those who thought that pandemic triage differed considerably (51%), and those who thought that it differed a little or not at all (46%). The main differences outlined in the open-ended answers related to the heightened importance of protective gear (respondents said it was more time-consuming, and it also limited their ability to carry out certain procedures—such as auscultation by ambulance service personnel), and the fact that triage in hospitals also took longer because special attention was paid to body temperature and breathing issues, which extended waiting times both for patients and ambulance crews. Respondents stressed the necessity of treating all patients as potentially infectious (ambulance work in Saaremaa in spring 2020 was described by one respondent as “bathing in corona”). Respondents emphasized heightened caution, consternation, and unfamiliarity, but there were also viewpoints maintaining that the danger was exaggerated. For example, more critical views towards the pandemic were also present: “unjustified abnormal attention to a rather ordinary disease” and “constant media coverage that created panic and stress in people”. Because COVID-19 infection control measures (e.g., paying attention to particular symptoms such as shortness of breath) applied to all patients in triage, our survey’s qualitative comment sections indicated that triaging tended to disadvantage those whose symptoms and complaints were linked to other diagnoses, raising issues relating to fairness and equal treatment.

Table 5 identifies the most important sources of stress during the first wave of the pandemic. Some statistically significant differences emerged between specialties: physicians were less likely to worry about lack of protective gear (*p* = 0.032) and about their own health (*p* = 0.005), and more likely to worry about potential resource scarcity (*p* = 0.001). Nurses were most concerned about their own mental health (*p* = 0.04), whereas EMTs were least concerned about workload (*p* = 0.001).

In addition to identifying up to five most important sources of stress from the questionnaire list, the respondents were also provided with a space to put forward any other concerns. The pandemic brought along overwhelming changes regarding the organization of both social and private lives, and this naturally created more stress. Worries ranged from childcare commitments to closed gyms, as well as to concerns that not all colleagues seemed to take the importance of personal protective equipment seriously enough. Fears were also raised about the potential for social stigmatization of medical professionals, and the stigma of the disease itself [13].

How should we prepare for future crises to decrease the stress on medical workers? The respondents highlighted that improved and clearer guidelines should help (59%), as would mental health and stress support for personnel (55%). A third of respondents thought improved communication within their organization (34%), as well as more general training on communicative skills to better converse with patients and their relatives (32%), were important. Better access to legal expertise/help was highlighted by 29%, 24% said training in ethical skills and knowledge was needed, and 20% pointed out a need for a greater involvement of ethics committees/ethics experts.

Finally, we asked whether the COVID-19 pandemic had changed HCP attitude towards their profession and work. We found that 81% agreed with the statement that the experience of the pandemic reconfirmed that their work mattered greatly. Some were also hesitant about confirming this (9.7%). Overall, no significant differences emerged between the three specialties here (*p* = 0.85), nor between age groups (*p* = 0.28). In contrast, 42% disagreed completely with the statement “I started doubting whether the chosen work suited me”, and 55% agreed with the statement that the pandemic did not, or mostly did not, change their attitude towards their work, although 21% thought that it did (or mostly did) change. Statistically significant differences emerged between specialties (*p* = 0.045), where physicians were more likely to disagree with the statement that the pandemic raised doubts about their chosen work (72% mostly or completely disagreed) while over half (58.33%) of the EMTs stated that they “could not say”. Given the chance, 10% of the respondents would have preferred not to work on the front lines, whereas 78% would have continued.

## 4. Discussion

Moral distress, anxiety, and depression in relation to COVID-19 have been widely studied amongst HCP [8,13,14,15,16], and several studies are available from nearby Latvia, Lithuania and Finland [17,18,19,20]. Significant levels of stress have been found in medical workers during COVID-19 using standard psychometric testing, where stress levels seem to correlate with the intensity of COVID-19 [6,21]. In terms of the major sources of stress, our survey data, which did not use standardized questionnaires, broadly correlated with those from a meta-study of qualitative research on COVID-19 impacts [22]. In Estonia, whilst the majority of respondents agreed that mental health and stress support for personnel (55%) would help in future crises, worries about one’s own mental health were highlighted by only 18% of the respondents, while seven other sources of stress were ranked as more important. In line with results from other countries, heightened caution, consternation, and unfamiliarity were underlined; but unlike other studies, hesitations and more critical views about the potentially exaggerated danger were also expressed, which might be related to the mild occurrence of the pandemic during the first wave [21,23]. Affirmation of the views of HCP that their work matters greatly, and that a large majority would not have given up their work in the front lines (even if they had been given a choice) are important findings amongst the abundance of literature that has focused on studying the increase in mental strain. These results could cautiously be interpreted as indicating the general resilience of healthcare professionals, with some even characterizing the first wave as “No worries nor stress—work as usual, with a few nuances”. On the other hand, our study was focused on the relatively mild first wave of the pandemic; research on the much more demanding second and third waves might reveal different results.

Estonian frontline HCP relied on their previous training and experience, and the majority of them reported confidence in their medical decision-making capabilities in the context of the pandemic. Sources of support for medical decision-making, as well as for more general support, are associated with various levels of the social-ecological model of wellbeing, ranging from individual self-care to public policy [24]. Operative hospital management communication and other organizational variables have been shown to impact and decrease psychosocial distress amongst HCP [25,26], and in Estonia we also found that institutional policies and guidelines were amongst the important factors supporting decision-making in this context.

Although the patient’s will is central to decision-making in the contemporary clinical context, in emergency medicine, triage, and generally resource-limited contexts, the situation might be more complex, as illustrated by Table 4. An important clinical criterion for treatment decisions in emergency contexts often includes prognosis [27,28], which was also prioritized amongst our respondents. In such contexts, the patient’s will plays an asymmetrical role—it is decisive in cases where the patient rejects treatment, but the desire for treatment (e.g., use of ventilation) could be limited due to resource scarcity or other reasonable limitations of treatment.

The ethics of resource allocation in a pandemic context was a much-debated topic at the beginning of the pandemic. Fairness regarding access to treatment was especially important from the perspectives of the elderly and the disabled populations. Many pandemic triage guidelines specifically excluded age as a relevant criterion [4,29]. Our study demonstrates the views of healthcare professionals on this question, where the expected outcome (prognosis) and the health status of the patient were prioritized before age, although certain correlations often do exist between all three of those criteria.

Study limitations pertain to potential selection bias (online survey). Nevertheless, the study highlights important insights into the impact of COVID-19 on ethical decision-making amongst frontline medical workers in the context of a small European country.

## 5. Conclusions

Overall, healthcare professionals in Estonia expressed confidence in their roles during the pandemic. More than half cited the complex ethical aspects of pandemic decision-making as their main source of doubt and uncertainty. In response to this uncertainty, respondents drew on their previous training and experience, the policies and guidelines of their institution, and support from their colleagues, to assist their decision-making during the pandemic. Although there was no consensus regarding whether pandemic triage was significantly different from normal triage, in the survey’s qualitative sections’ respondents expressed concern that triage procedures disadvantaged patients with symptoms or complaints unrelated to the coronavirus. Despite facing difficult decisions and significant stress during the first wave of the pandemic, most professionals agreed that their experience of the pandemic reconfirmed that their work mattered greatly.

## Figures and Tables

**Table 1 healthcare-10-00711-t001:** Participant age groups and specialties.

Specialty	
EMTs	15.60%
Physicians	21%
Nurses	59%
Age	
25–30 years	35%
31–42 years	30%
43–54 years	27%
55–60 years	8%

**Table 2 healthcare-10-00711-t002:** (Self-)confidence in clinical role during pandemic (Question 1) and (self-)confidence in case of medical resource scarcity (Question 2).

Confidence during the Pandemic								
	Physicians		EMTs		Nurses		All Specialties	
	Q1	Q2	Q1	Q2	Q1	Q2	Q1	Q2
Very confident	8.69%	6.50%	17.64%	2.90%	7.75%	3.80%	9.56%	4.30%
Mostly confident	50%	21.70%	47.05%	29.40%	47.28%	20.10%	47.84%	22%
Neither confident nor unconfident	30.43%	36.90%	23.52%	44.10%	34.88%	41%	32.05%	40.60%
Rather unconfident	8.69%	30.40%	2.94%	17.60%	8.52%	29.40%	7.65%	27.70%
Not at all confident	2.17%	4.30%	8.82%	5.80%	1.55%	5.40%	2.87%	5.20%
Q1: How (self)confident were you in your clinical role during the pandemic?								
Q2: How (self)confident were you in case medical resources would need to be limited?								

**Table 3 healthcare-10-00711-t003:** Factors that most supported respondents’ decision-making during the pandemic (up to 3 choices allowed).

Sources of Support for Medical Decision-Making			
	First Choice	Second Choice	Third Choice
My training and experience	37.11	25.5	28.4
Institutional policies	27.32	25.5	27.8
Collegial support	25.3	37.5	23.1
Management support	7.7	9.9	18.3

**Table 4 healthcare-10-00711-t004:** The views of frontline HCP on what criteria, in the context of resource scarcity, should form the basis for treatment decisions.

Criteria for Treatment Decisions			
	First Choice	Second Choice	Third Choice
Expected outcome	43.1	28.2	20
Health status of patient	32.8	38.2	14.2
Patient age	10.3	14.7	24.5
Patient’s will	6.3	9.4	18

**Table 5 healthcare-10-00711-t005:** Most important sources of stress during the 1st wave of pandemic (choice of up to 5).

Most Important Sources of Stress	
Worries about friends and family	57%
Rapidly changing pandemic situation in the country and the associated lack of knowledge	52%
Lack of knowledge/insufficient knowledge of COVID-19	52%
Worries about potential resource scarcity (personnel, medical equipment etc)	45%
Large workload	34%
Problems related to lack of appropriate protective gear	32%
Worries about one’s own physical health	32%
Also: worries about one’s own mental health (18%); not enough testing capacity (18%); communication problems and lack of information within an organization (13%); insecurities related to working in a new role (pre-pandemic employment outside of ICU) (7%).	

## Data Availability

The datasets used and analyzed during the current study are available from the corresponding author on reasonable request.

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
