# Peer review of "Frontline Healthcare Professionals’ Views Regarding the Impact of COVID-19 on Ethical Decision-Making: A Multicentre Mixed-Methods Study from Estonia"

_healthcare, 2022, doi:10.3390/healthcare10040711_

Round 1

Reviewer 1 Report

The paper entitled: “Frontline Healthcare Professionals’ Views regarding the Impact 2 of COVID-19 on Ethical Decision-Making. A Multicentre Study from Estonia” investigates frontline healthcare workers’ experiences and attitudes in relation to the COVID-19 pandemic’s ethical and psychosocial aspects 10 in Estonia. The authors surveyed 215 respondents from Oct 16th to Nov 30th, 2020. The sampling method was not probabilistic but purposeful. Therefore, is subject to potential selection bias. However, this paper examines a very important subject (ethical decision-making amongst medical workers who had a high risk of coming into contact with Covid-19 infected persons).

The research design and methods were clearly stated, findings are compelling, and the empirical resources are clearly presented. Finally, the conclusions summarized the results of the paper. On the other hand, references are updated.

The questionnaire contained both closed-ended and open-ended questions. However, the authors explore only the closed-ended questions. 116 respondents expressed open commentaries about the decision-making process. However, the authors do not examine in depth these commentaries.

The authors briefly examine the open commentary section in a few lines in p. 4. This analysis is quantitative and very shallow. The authors should fulfil a qualitative in-depth analysis of this section

Author Response

We thank the reviewers for their insightful and useful comments and criticisms and hope to have responded adequately in editing the new draft of the article. Below we respond to each question separately (changes in the article draft itself have been marked with red).

 Review 1

The authors briefly examine the open commentary section in a few lines in p. 4. This analysis is quantitative and very shallow. The authors should fulfil a qualitative in-depth analysis of this section.

We have added a description in the methods section. The open commentary sections followed particular questions in the survey (N=8) where the respondents could also answer “Other” and then explain further. We have now integrated qualitative insights into the discussions of those particular questions throughout the article with the aim of illustrating and elaborating the most significant findings of the quantitative sections. However, the main focus of the article is still on quantitative data analysis, as another already published article of ours focused on the qualitative analysis, linking qualitative data from the questionnaire with interviews (Kadi Lubi, Kadri Simm, Kaja Lempu, Jay Zameska & Angela Eensalu-Lind (2021): ‘Other patients become a secondary priority:’ perceptions of Estonian frontline healthcare professionals on the influence of COVID-19 on health (in)equality and ethical decision-making, Journal of Communication in Healthcare, DOI: 10.1080/17538068.2021.2013055)

Reviewer 2 Report

Dear authors, thank you for the opportunity to review this manuscript. The topic discussed is very important, but at the moment it needs to be refined before final publication. My main concerns are listed below.

The introduction is too laconic, it needs more information on what we already know about the mental health effects of the pandemic. Undoubtedly, the first period was the most turbulent - the literature already has a lot of information on this subject - it should be expanded what are the causes of the deterioration of the mental condition.
Additionally, what research hypotheses were formulated in the study? They should be presented and supported by the literature.
In the methodology section, please note the restrictions in place during the data collection period and provide the exact timeframe for data collection.
Chart 1 is illegible, proposes that the description of the study group should be presented in a table and only the most important information should be left in the description - do not duplicate it in the text.
Similarly with other questions and dependencies - presenting them in a table and listing only the most important things - definitely increases the clarity of the text and the ease of reading it.
There is no reference to world literature in the discussion, both from close neighbours (Lithuania, Latvia, Poland) and distant ones? How are your results in the international arena converging? Divergent? Were there population studies?
The conclusions are too long, it should be a short description.

Author Response

Responses to the reviewers regarding the manuscript “Frontline healthcare professionals' views regarding the impact of Covid-19 on ethical decision-making. A multicentre mixed-methods study from Estonia.”

We thank the reviewers for their insightful and useful comments and criticisms and hope to have responded adequately in editing the new draft of the article. Below we respond to each question separately (changes in the article draft itself have been marked with red).

Review 2

The introduction is too laconic, it needs more information on what we already know about the mental health effects of the pandemic. Undoubtedly, the first period was the most turbulent - the literature already has a lot of information on this subject - it should be expanded what are the causes of the deterioration of the mental condition. Additionally, what research hypotheses were formulated in the study? They should be presented and supported by the literature.

We have expanded the discussion in the introduction and also discussion sections. While we asked the respondents to identify sources of support as well as stress in the pandemic 1st wave context, the main focus of our study was on ethical decision-making in medical context and thus it encompassed also other foci (changes in pandemic triage, criteria for treatment decisions etc).Our method was to explore and identify rather than measure the issues and frontline healthcare workers attitudes and we have tried to clarify this focus better in the introductory sector.

In the methodology section, please note the restrictions in place during the data collection period and provide the exact timeframe for data collection.

Timeframe is described on p 4-5: A total of 215 respondents completed the survey during the data collection period from Oct 16th to Nov 30th, 2020. We have added information about the second wave and that no restrictions were in place during the time of the data collection.

Chart 1 is illegible, proposes that the description of the study group should be presented in a table and only the most important information should be left in the description - do not duplicate it in the text.

Similarly with other questions and dependencies - presenting them in a table and listing only the most important things - definitely increases the clarity of the text and the ease of reading it.
We have presented all figures now as tables and avoid duplications.      

There is no reference to world literature in the discussion, both from close neighbours (Lithuania, Latvia, Poland) and distant ones? How are your results in the international arena converging? Divergent? Were there population studies?

World literature and relevant results are included at the beginning of the “Discussion” section. As to nearby countries, we had results from Poland (Babicki et al) and have now also included relevant studies from elsewhere (Latvia, Lithuania, Finland). While there are many research articles that use standardized questionnaires for measuring symptoms of anxiety and depression for healthcare workers during the pandemic, our study employed a different method and a slightly different focus, thus it is not always directly comparable. Yet we believe that it provides affirmation of healthcare professionals’ views that their work matters greatly and more than ¾ would not have given up their work in the frontlines even if they would have been given a choice (despite the increased stress and mental strain). Also, our study demonstrates that it is the ethical aspects of decision-making that were the main source of doubt. We find these to be important findings amongst the abundance of literature that focused on measuring stress and anxiety.

The conclusions are too long, it should be a short description.

We have shortened the conclusion.

Thank you!

Kadri Simm

Jay Zameska

Kadi Lubi

Round 2

Reviewer 2 Report

The authors addressed most of the concerns previously reported. The article meets the publication criteria.